# Epidemiology, Pathogenesis, and Diagnostic Strategy of Diabetic Liver Disease in Japan

**DOI:** 10.3390/ijms21124337

**Published:** 2020-06-18

**Authors:** Yoshio Sumida, Toshihide Shima, Yasuhide Mitsumoto, Takafumi Katayama, Atsushi Umemura, Kanji Yamaguchi, Yoshito Itoh, Masashi Yoneda, Takeshi Okanoue

**Affiliations:** 1Division of Hepatology and Pancreatology, Department of Internal Medicine, Aichi Medical University, Nagakute, Aichi 480-1195, Japan; yoneda@aichi-med-u.ac.jp; 2Center of Gastroenterology and Hepatology, Saiseikai Suita Hospital, Osaka 564-0013, Japan; shima0301d@suita.saiseikai.or.jp (T.S.); mitsumoto-gi@umin.ac.jp (Y.M.); t-kata@koto.kpu-m.ac.jp (T.K.); okanoue@suita.saiseikai.or.jp (T.O.); 3Department of Gastroenterology and Hepatology, Graduate School of Medicine, Kyoto Prefectural University of Medicine, Kyoto 602-8566, Japan; aumemura@koto.kpu-m.ac.jp (A.U.); ykanji@koto.kpu-m.ac.jp (K.Y.); yitoh@koto.kpu-m.ac.jp (Y.I.)

**Keywords:** hepatic fibrosis, type IV collagen 7s, hepatocellular carcinoma, patatin-like phospholipase domain containing 3

## Abstract

Type 2 diabetes (T2D) is closely associated with nonalcoholic fatty liver disease (NAFLD). Nonalcoholic steatohepatitis (NASH), a severe form of NAFLD, can lead to cirrhosis, hepatocellular carcinoma (HCC), and hepatic decompensation. Patients with T2D have twice the risk of HCC incidence compared with those without T2D. Because the hepatic fibrosis grade is the main determinant of mortality in patients with NAFLD, identifying patients with advanced fibrosis using non-invasive tests (NITs) or imaging modalities is crucial. Globally, the fibrosis-4 index (FIB-4 index), NAFLD fibrosis score, and enhanced liver fibrosis test have been established to evaluate hepatic fibrosis. Two-step algorithms using FIB-4 index as first triaging tool are globally accepted. It remains unknown which kinds of NITs or elastography are best as the second step tool. In Japan, type IV collagen 7s or the CA-fibrosis index (comprising type IV collagen 7s and aspartate aminotransferase (AST)) is believed to precisely predict advanced fibrosis in NAFLD. Patients with NAFLD who have high non-invasive test results should be screened for HCC or esophageal varices. Risk factors of rapid fibrosis progression in NAFLD includes age, severe obesity, presence of T2D, menopause in women, and a patatin-like phospholipase domain containing the 3 GG genotype. Patients with NAFLD who have these risk factors should be intensively treated with lifestyle modification or pharmacotherapies for preventing liver-related mortality.

## 1. Introduction

Obesity and type 2 diabetes (T2D) are major risk factors in the development of nonalcoholic fatty liver disease (NAFLD). NAFLD is histologically categorized as simple steatosis (nonalcoholic fatty liver: NAFL) and may progress to nonalcoholic steatohepatitis (NASH), characterized by hepatocytes degeneration and inflammation. The latter can advance to fibrosis, cirrhosis and hepatocellular carcinoma (HCC). Patients with T2D not only are at a high risk of developing NAFLD but also display a progression toward significant liver fibrosis that is double that observed in other patients with NAFLD [1,2]. Insulin resistance has been considered an important factor in the development and progression of NAFLD [3]. However, the determinants of this increased risk in patients with T2D are still unclear [4]. Obesity, T2D, hypertension and dyslipidemia have been observed in NAFLD, and higher rates of overall mortality, mortality related to liver and cardiovascular diseases (CVD) have been reported when T2D and NAFLD were present concomitantly [5,6]. The leading cause of mortality in NAFLD patients is CVD [7]. NAFLD is an independent risk factor of coronary sclerosis [8], atrial fibrillation [9], and left ventricular dysfunction [10,11]. In daily clinical practice, we should pay attention to CVD event and control other risk factors, such as hypertension, dyslipidemia, and T2D.

With the Westernization of the Japanese lifestyle, public interest in lifestyle-related diseases has increased rapidly. In 2007, the Japan Society of Diabetes Mellitus reported that among the causes of death for 18,385 individuals with diabetes, liver cancer was the leading cause (8.6%), while death from liver cirrhosis also was very common (4.7%), 13.3% of deaths among patients with diabetes were attributable to liver disease [12]; however, the prevalence of hepatitis virus infections and heavy drinking were not analyzed in this study. According to a nationwide survey (2001–2010) in Japan, liver-related disease (HCC and cirrhosis) is the third leading cause of mortality (9.3%) in T2D [13]. Most Japanese people are not as obese as Caucasians, but NAFLD is becoming more common in Japan (24.6%–34.7% of the adult population) [14,15,16]. This phenomenon may be explained by higher prevalence of patatin-like phospholipase domain containing 3 (*PNPLA3*) G allele in Asia compared with western countries [15,17]. At present, NASH is one of the most important liver diseases in many countries, including Japan. Oxidative stress has a major role in the pathogenesis of NASH and fibrosis progression [18]. As we previously reported, serum thioredoxin (TRX) levels, an indicator of oxidative stress, are higher in patients with NASH than those with NAFL [19]. TRX levels were positively correlated with fibrosis stage in NASH. Several antioxidative agents are therapeutic options for NASH, including vitamin E [20,21], glutathione [22], probucol [23], and pentoxifylline [24].

In 2008, the Japan NASH Study Group (of which Takeshi Okanoue was a representative), which was supported by the Ministry of Labor and Welfare, started to study NAFLD/NASH. The purpose was to elucidate the epidemiology, pathophysiology, genetic backgrounds and long-term prognosis of NAFLD, and to develop noninvasive tests (NITs) for diagnosis of NASH.

In this review article we describe the present status of liver disease in people with diabetes in many countries, including Japan.

## 2. Epidemiology of Fatty Liver

The prevalence of NAFLD differs by ethnic group and diagnostic tools, ranging from 15% to 70% [25]. The worldwide prevalence of NAFLD is about 30–35% [17]. Liver biopsy remains the “gold standard” for diagnosis of necro-inflammatory activity and location of fibrosis although it is an invasive procedure. Most epidemiological studies in NAFLD were identified using ultrasonography (US) of the liver or a combined US and serum liver test (alanine aminotransferase: ALT). However, US is sometimes unable to detect less than 30% liver fat content [26,27]. Serum ALT level frequently shows normal range, especially in advanced staged NASH (known as “burned-out NASH”). Thus, at present, the exact epidemiology of NAFLD with/without T2D is uncertain. Approximately 50–60% of T2D patients may have NAFLD [6,28,29,30].

The Japan Society of Ningen Dock (a health check-up organization) reported in 2008 that the prevalence of liver dysfunction, including fatty liver, was 31.9% in men and 17.1% in women, based on a study carried out of 1,814,864 adult men and 1,236,903 adult women [31]. Kojima et al. reported that the prevalence of fatty liver detected by medical health checks increased year after year, from 12.6% in 1989 to 30.3% in 1998 [32]. The majority of fatty liver disease comprises alcoholic fatty liver and NAFLD, including NASH. Hamaguchi et al. reported that the prevalence of NAFLD was 23.3% in Japanese adults [26]. There is a sex difference in the incidence of NAFLD; men are more likely than women to develop fatty liver. Racially, NAFLD prevalence in Asians is never lower than that in Caucasians. *PNPLA3* genotype has been shown to play a significant role in NAFLD pathogenesis, progression of liver fibrosis and even in the development of HCC [15,17,33,34]. In the Japanese population, the prevalence of the risk allele in *PNPLA3* is high, which might be one reason in this point [17,33]. There is also a sex difference in the age distribution of NAFLD. In our study the incidence of fatty liver remained unchanged for men from the 30s to the 60s, whereas in women, the prevalence of fatty liver increased gradually with age and, when women were in their 60s and beyond, the prevalence reached a higher level than that of men.

## 3. Prevalence of NAFLD in Patients with Diabetes/Impaired Glucose Tolerance

NAFLD prevalence differs by glycemic status. Kojima et al. reported that the prevalence of fatty liver was 18.6% in subjects with normal glucose metabolism (fasting blood sugar (FBS) < 110 mg/dl), 43.7% in borderline subjects (110 mg/dl ≤ FBS <126 mg/dl), and 53.3% patients with diabetes (FBS ≥ 126 mg/dl). FBS ≥ 110 mg/dl was an independent risk factor for fatty liver [32]. In 8352 Japanese patients who received health check-ups from 2009 to 2010, the incidence of NAFLD was 25.6% in those with normal fasting glucose, 56.2% in those with impaired fasting glucose (IFG), and 68% in those with T2D [14]. The overall prevalence of NAFLD in Southeast Asia varies from 9% to 45% in the general population and 6%–62% in patients with T2D [35]. As described above, the diagnostic tools for the detection of NAFLD in individual studies is not uniform. In the National Health and Nutrition Examination Study in Japan, conducted in 2007, 8,900,000 (of the total Japanese population of around 130,000,000) people were strongly suspected of having diabetes; the number of people with an undeniable possibility of having diabetes was 13,200,000, and the number of people with some possibility of having diabetes was 22,100,000, which was 1.6-fold higher than 10 years earlier [36]. NAFLD often precedes the development of T2D [37]. Recently it has been considered that NAFLD is an early predictor and determinant for development of diabetes. NAFLD has been shown to increase the risk of diabetes by 1.6 to 6.8 times in meta-analyses [38,39]. Miyaaki et al. examined the relationship between the stage of hepatic fibrosis and the prevalence of diabetes in Japanese patients. In the mild fibrosis group, 42% were complicated with diabetes, but in the severe fibrosis (bridging fibrosis or cirrhosis) group, the prevalence was as high as 71% (*p* = 0.02). Diabetes might be an important factor responsible for the development of hepatic fibrosis in NAFLD [40].

We reported the results of the rate of abnormal liver function tests in 5642 (male: 3164, female: 2346) Japanese patients with T2D (average age; men: 62.2 y, women: 64.8 y) in which alcohol consumption and frequency of hepatitis B (HBV) and C virus (HCV) infection were analyzed [41]. The positive rate of HBsAg was 1.7% (male: 1.8%, female: 1.6%) and of anti HCV Ab was 5.1% (male: 5.1%, female: 5.0%); however, most patients who were HBsAg positive showed very low levels of serum HBV DNA (< 2.6 log IU/mL) and 38% of the patients who were anti-HCV positive were negative for HCV RNA. The frequencies of habitual drinkers (daily alcohol intake > 60 g for more than 5 years) were 6.9% in male and 0.9% in female. Among the patients with NAFLD, the frequency of NASH and advanced stage NASH was significantly higher in male patients with T2D than in male patients without T2D. From these results, we suspect that in more than 80% of patients in Japan with diabetes who have liver injury, the reason for the liver injury is NAFLD. The incidence of HCC was 1.8% (*n* = 48) in males and 0.9% (*n* = 19) in females.

## 4. Risk of Cancer in Patients with NAFLD and Diabetes

NAFLD is a slowly progressive disease. The progression of one fibrosis stage is every 14 years in NAFLD and every 7 years in NASH; however, in 20% of NASH cases, fibrosis progresses more rapidly [42]. The coexistence of NAFLD and T2D worsens the course of both T2D and NAFLD [43]. A few decades ago, it was established that there is a substantially increased risk of mortality from cirrhosis and HCC in patients with T2D [44,45]. One report showed that patients with diabetes had an approximately threefold higher risk of dying of chronic liver diseases (CLDs), mainly associated with NAFLD [46]. Multiple studies and meta-analyses have claimed that T2D is associated with an increased risk of cancer at several sites, including liver, pancreas, endometrium, colorectum, breast, bladder, and a decreased risk of prostate cancer [47].

Recently we reported a follow-up study (average of 4.5 years) of 3999 patients with T2D [48]. Expected deaths in patients with T2D were estimated using age-specific mortality rates in the general population of Japan. The all-cancer mortality was significantly higher in patients with T2D than in the general population (standardized mortality rate (SMR) 1.58, 95% confidence interval (CI) 1.33–1.87). Among malignancies, HCC conferred the highest mortality risk in patients with T2D (SMR 3.57, 95% CI 2.41–5.10). HCC-associated mortality risk in patients with T2D remained significantly high (SMR 2.56, 95% CI 1.64–3.97) after adjusting for high positivity rates of hepatitis B surface antigen (1.7%) and anti-hepatitis C virus (5.3%). In patients with T2D who have low platelet counts (< 200 × 10^3^/μL), the SMR of HCC increased from 3.57 to 6.58 (95% CI 4.34–9.58). Patients with T2D who have a platelet count > 200 × 10^3^/μL showed no increase in mortality risk (SMR 0.68) of HCC. Our study is the first to demonstrate the significantly higher mortality of HCC in patients with T2D compared with the general population, even after adjusting for hepatitis virus positivity.

A Chinese cohort study showed that T2D was associated with an increased incidence of non-B, non-C HCC [49]. Patients with T2D are at high risk of NASH with liver fibrosis and are more likely to develop NASH-related cirrhosis and eventually NASH-related HCC [2,50,51,52]. The second cause was the increased HCC-associated mortality risk in patients with T2D who also have HBV or HCV hepatitis, especially in China, Taiwan and Korea. A study in Taiwan reported that T2D was significantly associated with an increased incidence of HCC in patients with HCV infection (risk ratio (RR), 3.52) and in HBV carriers (RR 2.27) [53]. A meta-analysis further supports the finding that T2D is a risk factor of HCC in patients with HCV infection (RR 1.90) and those with HBV infection (RR 1.69) [54]. Therefore, T2D might accelerate mortality in NASH-related and virus-associated HCC.

Patients with NASH who have advanced fibrosis are at a high risk of HCC [55,56]. The influence of insulin treatment on cancer risk is controversial. Although mechanisms of oncogenesis include insulin resistance, hyperinsulinemia, hyperglycemia, and inflammation [47,48], previous studies suggested hyperinsulinemia as a central player in the association between T2D and cancer development and progression [57,58]. Exogenous insulin administration might be implicated in the pathogenesis of increased cancer risk [59]. Here the mortality rate of all cancers was comparable between insulin and no-medication groups. However, a more precise analysis of the type, dose, and duration of insulin is necessary for discussing this agenda. The JDS/Japanese Cancer Association Joint Committee on Diabetes and Cancer reported that there was limited evidence as to whether any particular antidiabetic agent might influence cancer risk [60]. The relationship between glycemic control and cancer risk has also been controversial. Here the all-cancer mortality was higher in patients with T2D who have low glycated hemoglobin (HbA1c) (<7.0%) than in those with high HbA1c (≥7.0%) at the time of enrollment. Because HbA1c levels in the patient cohort were not monitored during the follow-up period, this result should be carefully interpreted. A meta-analysis reported that there was no evidence to support the hypothesis that improved glycemic control reduces the risk of cancer incidence or mortality [61,62].

## 5. The PNPL3 SNP and HCC- or Liver-Related Mortality

The PNPLA3 GG genotype seems to have an approximately twofold higher risk of developing hepatic steatosis, a three- to four-fold higher risk of developing NASH and cirrhosis, and an up to 12-fold higher risk of developing HCC as compared with the PNPL3 CC genotype [63]. In 110,761 individuals from the Danish general population and 334,691 individuals from the UK Biobank, the combination of PNPLA3, transmembrane 6, superfamily member 2 (TM6SF2) and hydroxysteroid 17-beta dehydrogenase 13 (HSD17B13) is associated with an up to 12-fold higher risk of developing cirrhosis and an up to 29-fold higher risk of developing HCC [64]. According to the U.S. National Health and Nutrition Examination Survey (NHANES), 1988–1994, with 27 years of linked mortality data, the PNPLA3 GG genotype had an 18-fold higher risk of developing liver-related mortality [65]. In previous studies, the PNPLA3 SNP did not play any role in obesity, insulin resistance, or incidence of T2D.

In Japan, we previously reported that severe hepatic fibrosis (F3/4) and PNPLA3 GG are significant factors associated with incident HCC development in Japanese patients with biopsy proven NAFLD [56]. A multicenter study from Japan by Korenaga and colleagues demonstrated that the SNPs of PNPLA3, when juxtaposed with another zinc finger protein 1 (JAZF1), were associated with development of HCC in patients with T2D who did not have hepatitis virus infection [66]. In addition, we newly identified rs17007417 in dysferlin (*p* = 5.2 × 10^−7^, OR (95% CI) = 2.74 (1.84–4.06)) as an SNP-associated with NASH–HCC [33] (Table 1). However, external validation studies are required to confirm this result.

## 6. Noninvasive Tests for Diagnosis of NAFLD and Staging Fibrosis

### 6.1. Evaluation of Hepatic Steatosis

US is the most common means for detecting hepatic steatosis; however, its sensitivity has been determined to be 85% [67] and its detection rate is lower if the hepatic steatosis is less than 30% [68]. Innovative imaging technologies for detecting mild steatosis (>5% of hepatocytes affected) have been developed, such as the vibration-controlled transient elastography (VCTE) controlled attenuation parameter (CAP), magnetic resonance imaging–proton density fat fraction (MRI-PDFF), and the ultrasound-guided attenuation parameter (UGAP) [69]. NITs for predicting steatosis in NAFLD have also been proposed to avoid imaging tests or biopsies (Table 2). The fatty liver index (FLI) [70], the SteatoTest [71] and the NAFLD liver fat score [72] reliably predict steatosis for severely obese people [73]. However, these biomarkers are not suitable for Asians because obesity is defined as a BMI >25 kg/m^2^ in Asian people but >30 kg/m^2^ in Western people. There are no validation studies of these NITs from Japan as so far, because US diagnosis is routinely performed by medical health check-up or general physicians in Japan. The FLI was validated by a study of 13,122 subjects in western China [74]. In Japan, the FLI is a useful predictor of incident diabetes [75].

A healthy ALT level is defined as 19 to 25 U/L for women and 29 to 33 U/L for men [77,78]. In our study with Japanese participants, these were 11 to 21 U/L for women and 13 to 24 U/L for men, and healthy people show an AST > ALT within the normal range of AST and ALT. Thus, we consider that people with >25 U/L of ALT in women, >30 U/L of ALT in men and/or AST<ALT within the normal range of ALT with a BMI of >25 kg/m^2^ are candidates for receiving NAFLD screening.

### 6.2. Evaluation of Hepatic Fibrosis (Table 3)

Advanced hepatic fibrosis is the most important risk factor for not only incident HCC but also liver-related mortality in NAFLD [79]. In patients with T2D, the prevalence of advanced fibrosis (F3/4) is estimated to be 17% (95% CI 7.2%–34.8%) by liver biopsy [28,80], 7.3%–24.9% by VCTE and 4.3%–7.1% by magnetic resonance elastography (MRE) [81]. All institutions do not have these innovative modalities such as VCTE or MRE. Therefore, it is important to establish simple, sensitive and low-cost NITs for the screening and staging of fibrosis in epidemiological study and in practice [82,83]. A reliable biochemical marker for advanced stages of NASH has not been established. Among a variety of NITs (Table 3), the fibrosis-4 index (FIB-4 index) and NAFLD fibrosis score (NFS) are the most appropriate to identify advanced fibrosis (F3/4) for general physicians (as first step) because these scoring systems consist of common parameters in daily clinical practice. Vilar-Gomez et al. reported that NFS and the FIB-4 index are useful screening tools for determining the stage of liver fibrosis to be routinely applied in clinical practice [84]. Thus, the FIB-4 index and NFS are now recommended for excluding advanced fibrosis (F3/4) in the American Association for the Study of Liver Diseases (AASLD) practice guidance 2018 [85]. NFS was calculated by the formula: 1.675 + 0.037 × age (years) + 0.094 × BMI (kg/m^2^) + 1.13 × impaired fasting glucose/diabetes (yes = 1, no = 0) + 0.99 × AST/ALT ratio − 0.013 × platelet count (×10^9^/L) − 0.66 × albumin (g/dL) [86]. NFS is useful but too complex, because it requires six parameters. In contrast, the FIB-4 index has similar accuracy to the NFS and has only four parameters [87,88]. The FIB-4 index must be the most simple and cheapest test to define the stage of liver fibrosis. However, the FIB-4 index has also several drawbacks. First, low and high cutoff points are variable according to ethnics. Low cut-off value of FIB-4 index was generally accepted as 1.3 in Western countries [87], while it was 1.45 in Asia [88,89]. Second, FIB-4 index requires an intermediate group. NAFLD patients classified into that group have to receive other NITs or liver biopsies. Third, it is a concern that FIB-4 index may overpredict fibrosis in older patients [90,91], because its formula includes age. On the basis of data in JSG-NAFLD, the new proposed low cutoff points are 1.88 in 60–69 years, and 1.95 in > 70 years [91]. Moreover, we found that FIB-4 index might be inferior in NAFLD patients with T2D compared to those without T2D [92]. Although its precise mechanism of inferiority in T2D patients remains unknown, platelet count tends to be higher in NAFLD patients with T2D compared to those without T2D. FIB-4 index in NAFLD patients with T2D is lower than in those without T2D at the same fibrosis stages. Fourth, Shah S and colleagues feel that a low cut-off of 1.3 may be inappropriate, as it would include patients with F2 fibrosis [93]. F2 fibrosis confers an increased mortality of liver-related diseases compared with no fibrosis (F0) (HR: 2.52) [79]. “Active NASH” which requires intensive treatment is defined as NASH with NAFLD activity score (NAS) ≥ 4 and ≥ F2. FAST score, consisting of three parameters including VCTE-based CAP, VCTE-based LSM, and AST, can predict “active NASH” [94,95]. However, the FIB-4 index is believed to be sufficient as a triaging tool to exclude advanced fibrosis (F3/F4).

Thus, we recently developed markers that were reliable (for screening NASH: the FM-NASH index; for staging liver fibrosis: the FM-fibro index) [96] and simple (for screening NASH: the CA index-NASH; for staging liver fibrosis: the CA index-fibrosis) [97] to screen NASH and to estimate advanced liver fibrosis in NASH. We consider that both the FM index and CA index are superior to other NITs. Especially, the CA index is very simple and highly accurate; it is composed of type IV collagen 7s and AST. Calculation of the CA index is shown as CA index-NASH = 0.994 × (type IV collagen 7s (ng/mL)) + 0.0255 × (AST (IU/l)); and CA index-fibrosis = 1.5 × (type IV collagen 7s (ng/mL)) + 0.0264 × (AST (IU/l)) (Table 3). The CA index NASH, consisting of type IV collagen 7s and AST, showed high AUROC (training: 0.857; validation: 0.769) [98] for differentiating between NAFL and NASH. We consider that type IV collagen 7s is a central fibrosis marker in NASH. Type IV collagen 7s is a fragment of collagen type IV, and it is part of the extracellular matrix, which forms the basement membrane. Type IV collagen 7s is an established biochemical marker of liver fibrosis. An increase in serum levels of type IV collagen 7s is accompanied by progression of liver fibrosis in various chronic types of hepatitis, including NASH. Additionally, the efficacy for diagnosis of advanced fibrosis in NASH has been established. A previous study showed the efficacy of type IV collagen 7s for diagnosing NASH or advanced fibrosis of NAFLD [99,100,101]. Type IV collagen 7s has been estimated by radioimmunoassay in Japan; however, from April 2020, it was able to be estimated by ELISA in Japan. A platelet count is a simpler marker for screening patients with T2D who are at high risk of HCC. Kawamura et al. identified a platelet count <150 × 10^3^/μL as a risk factor of HCC in patients with NAFLD (HR 7.19) [55]. Detection of patients with advanced hepatic fibrosis by monitoring platelet counts may be effective for early identification of those at high risk of HCC. A high FIB-4 index score >2.67 is also a useful marker for HCC risk (RR 14.0). Formula, strengths, and weakness in each NIT are summarized in Table 3.

In conclusion, the FIB-4 index is believed to be sufficient for general physicians or endocrinologists as a triaging tool to exclude advanced fibrosis (F3/4). The CA-fibrosis and enhanced liver fibrosis (ELF) tests are useful for hepatologists (as a second step) to consider performing liver biopsy. Recently, two studies from Europe [102] and the US [103] demonstrated that VCTE controlled CAP and the liver stiffness measurement (LSM) in assessing steatosis and fibrosis in patients with suspected NAFLD. They found that CAP and LSM by FibroScan to assess liver steatosis and fibrosis with AUROC values ranged from 0.70 to 0.89 [84]; however, VCTE was less accurate in distinguishing a lower fibrosis stage, higher steatosis grades, or the presence of NASH [103]. In a few Japanese studies, 9.6–10.8 kPa of LSM was an optimal cutoff value for identifying advanced fibrosis in NAFLD [104,105,106].

### 6.3. Combination of NITs and Imaging Modalities

Recently, two systematic review articles covering the diagnosis of NASH and its staging of liver fibrosis were reported from the US [107] and Europe [108] in which Younossi et al. recommended a combination of biochemical NITs and MRE for the best predictive performance. We recently reported the usefulness of the combination of LSM by VCTE and NITs for the diagnosis of the stage of liver fibrosis in NASH. VCTE has a slightly limited applicability for patients with NAFLD; however, we demonstrated that using concurrent measurement with certain biomarkers to predict advanced liver fibrosis (> F3/4), such as the FM-fibro index (AUROC; 0.945), type IV collagen 7s (AUROC; 0.925), FIB-4 index (AUROC; 0.927) and CA fibro index (AUROC; 0.919), significantly improved the diagnostic accuracy [109]. We should establish the best combination of NITs and imaging studies from the perspective of the cost–benefit balance.

## 7. Two-Step Algorithm in NAFLD Diagnosis

For diagnosing NAFLD, a two-step algorithm is now globally accepted [84,89,110,111], as shown in Figure 1. As the first step, patients with NAFLD who have a FIB-4 index <1.3 who are unlikely to have advanced stage (F3/4) can be followed up. The FIB-4 index should be measured every 2 or 3 years. As the second step, patients with NAFLD who have a FIB-4 index ≥1.3 are recommended to receive VCTE. Patients with VCTE ≥10 kPa have to undergo liver biopsy unless they have any contra-indications for performing liver biopsy, and subjects with VCTE ≤10 kPa can be followed up with annual VCTE. One concern is that not all institutions have VCTE even in Japan. However, there are alternative NITs for evaluating hepatic fibrosis, including the type IV collagen 7s, CA-fibrosis index and ELF test. MR elastography is positioned as third step, considering its cost, duration, and contraindication in patients with pacemakers or claustrophobia. Better algorithms are expected to be developed from the point of view of accuracy, non-invasiveness, and cost–benefit balance owing to the huge number of people with NAFLD. Because the FIB-4 index is influenced by age, some reports have suggested a low cutoff value of 2.0 in older patients (aged 65 to 70 y or older) [90,91]. Old NAFLD patients showing FIB-4 index <2.0 are unlikely to have advanced fibrosis.

It remains to be resolved which is the best modality as the second step among a variety of NITs. VCTE is recommended as the second step, considering its simplicity, portability, and non-invasiveness. In a few Japanese studies, 9.6–10.8 kPa of LSM was an optimal cutoff value for identifying advanced fibrosis in NAFLD [104,105,106]. A two-step referral pathway system from Canada suggested that the cutoff value of VCTE was 8.0 kPa after triaging by FIB-4 index [111]. An international collaboration study showed that the optimal threshold of VCTE for detection of advanced fibrosis (F3/F4) was 8.8 kPa [112]. Thus, the cutoff values of VCTE for identifying advanced fibrosis varied from 8.0 to 10.8 kPa. In this two-step algorithm, 10.0 kPa was adapted as the cutoff values of VCTE for identifying advanced fibrosis. However, this two-step algorithm should be validated by international, multi-center, and prospective trials. Other candidates as the second step modality include type IV collagen 7S, CA-fibrosis index, Mac-2 binding protein glycan isomer (M2BPGi) [113] and ELF test [114]. M2BPGi, a novel hepatic fibrosis marker, has been developed in Japan. M2BPGi can predict fibrosis stage in CLD, including NAFLD [115,116,117]. In the Korean population with CLD, sequential combination of FIB-4 index followed by M2BPGi can identify advanced fibrosis (≥F3) evaluated by MRE [113]. We are now planning to validate this two-step algorithm using FIB-4 index and M2BPGi in Japanese population with biopsy-proven NAFLD.

The enhanced liver fibrosis (ELF) test is a non-invasive blood test that measures three direct markers of fibrosis: hyaluronic acid (HA), procollagen III amino-terminal peptide (PIIINP), and tissue inhibitor of matrix metalloproteinase 1 (TIMP-1) [98]. According to a two-step algorithm from EU [113], ELF test can be applied to the intermediate group of FIB-4 index (1.3–3.25). If NAFLD patients have an ELF score of 10.35 or above, they are likely to have advanced fibrosis [113]. Recently, the usefulness of ELF test was validated in a Japanese NAFLD population [118]. Combinations or sequential procedures using VCTE complement the diagnostic performance of ELF testing for the identification of advanced fibrosis. On the view of economic cost, the combination of FIB-4 index plus ELF test is superior to the combination of FIB-4 index plus VCTE [119]. Although ELF test will be covered by health insurance in Japan around 2021, most hepatologists have little knowledge about ELF tests. External validation studies in an international multi-institutional setting are required to confirm this algorithm as shown in Figure 1. It remains unknown whether the genetic variant PNPLA3 I148M may complement these NITs for NAFLD surveillance.

Recently, we constructed artificial intelligent (AI) systems for screening NAFLD and staging fibrosis. These systems might be very useful in clinical practice, including for medical health check-up programs and epidemiological studies of NAFLD, and they certainly significantly reduce medical costs in clinical medicine related to NAFLD. We are now planning to construct these AI systems using Western patients with NAFLD.

## 8. Risk Factors of Rapid Fibrosis Progression in NAFLD

A notable minority of patients with NAFLD progress to more advanced disease that is characterized by NASH and subsequent fibrosis and cirrhosis. To date, there are few high-quality prospective data on the progression of NAFLD. Risk factors of rapid fibrosis progression in NAFLD includes the existence of NASH [120], age [2,120], severe obesity [117,121,122], high fructose consumption [123], insulin resistance [3], presence of T2D [2,17], high HbA1c levels [124], menopause in women [125,126], high ALT levels [127,128], and PNPLA3 G allele [129]. NAFLD progression is influenced by a combination of genetic and environmental factors.

## 9. Message to General Physicians for Avoiding Misdiagnosis of NASH with Severe Fibrosis

First of all, general physicians determine AST, ALT, and platelet count for calculating FIB-4 index in NAFLD patients. Auto-calculator is uploaded in Japan Society of Hepatology (https://www.eapharma.co.jp/medicalexpert/product/livact/fib-4/calculator.html). NAFLD patients with a low FIB-4 index score can be followed up. However, coexisting metabolic diseases (T2D, dyslipidemia, and hypertension) should be appropriately treated in those patients. FIB-4 index can be determined every 2 or 3 years. NAFLD patients with intermediate FIB-4 index scores should be considered to perform VCTE or refer to hepatologists. NAFLD patients with a high FIB-4 index should be immediately referred to hepatologists for screening of HCC and esophageal varices. Two-step algorithm using the FIB-4 index as the first step for excluding advanced fibrosis in primary care could reduce unnecessary overreferral and increase case-finding of patients at risk of complications of liver disease. Three parameters including ALT, body weight, and HbA1c (ABC) should be strictly controlled for prevention of fibrosis progression in NAFLD [130].

## 10. Conclusions

Recently, NAFLD has received much attention in many countries, including Japan. Improving NITs or imaging modalities for screening NAFLD/NASH and staging liver fibrosis in NASH might clarify the epidemiology and prognosis of fatty liver in patients with diabetes. Furthermore, the genetic variant PNPLA3 I148M may complement these NITs for NAFLD surveillance [33,65]. Improving survival in patients with NAFLD requires collaboration between hepatologists and general physicians or diabetes specialists.

## Figures and Tables

**Figure 1 ijms-21-04337-f001:**
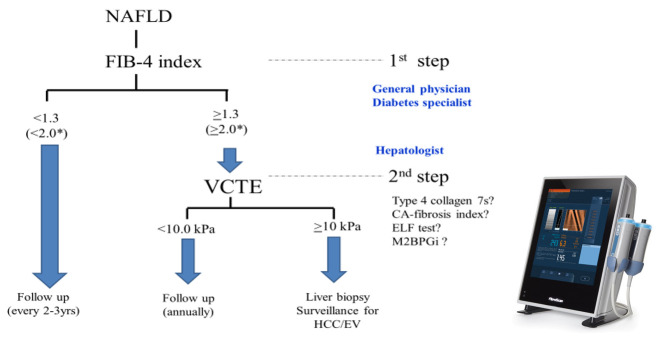
Two-step algorithm for determining NAFLD using the FIB-4 index (first step) and vibration-controlled transient elastography (VCTE) (second step). * older patients (aged 60 to 70 years or older); NAFLD: nonalcoholic fatty liver disease, FIB-4: fibrosis-4, VCTE: vibration-controlled transient elastography, ELF: enhanced liver fibrosis, M2BPGi: Mac-2 binding protein glycan isomer, HCC hepatocellular carcinoma, EV: esophageal varices.

**Table 1 ijms-21-04337-t001:** The distribution of SNPs in patatin-like phospholipase domain containing 3 (PNPLA3) and dysferlin in 58 Japanese patients with nonalcoholic steatohepatitis (NASH)–hepatocellular carcinoma (HCC) [33].

	**PNPLA3**				**Total**
		Major allele (TT) 29.4% *	Hetero (TG) 4% *	Risk allele (GG) 21.1% *	
DYSF	Major allele (CC) 68.5% *	1 (1.7%)	7 (12.1%)	17 (29.3%)	43%
	Hetero (CT) 28.8% *	4 (6.9%)	10 (17.2%)	11 (19.0%)	43%
	Risk allele (TT) 2.6% *	1 (1.7%)	1 (1.7%)	6 (10.3%)	13.7%
Total		10.3%	31.0%	58.6%	

* prevalence of the general population; PNPLA3: patatin-like phospholipase domain containing 3, DYSF: dysferlin.

**Table 2 ijms-21-04337-t002:** Non-invasive tests (NITs) for predicting steatosis in hepatic steatosis.

Index	Author (Nation)	Publication (Year)	Parameters
Fatty liver index (FLI)	Bedogni (Italy)	BMC Gastroenterology (2006) [70]	BMI, WC, TG, γGTP
NAFLD liver fat score	Kontronen (Finland)	Gastroenterology (2009) [72]	MetS, T2D, IRI, AST, AST/ALT ratio
Hepatic steatosis index (HSI)	Lee (Korea)	Dig Liver Dis (2010) [76]	8 × AST/ALT ratio + BMI + (female: + 2, diabetes: + 2)
SteatoTest (ST)	Poynaud (France)	Comp Hepatol (2005) [71]	ALT, α2-macroglobulin, apolipoprotein A-I, haptoglobin, total bilirubin, γGTP, cholesterol, TG, glucose, age, sex, BMI

BMI: body mass index, WC: waist circumference, TG: triglyceride, γGTP: gamma glutamyl transpeptidase, MetS: metabolic syndrome, T2D: type 2 diabetes, IRI: immune-reactive insulin, AST: aspartate aminotransferase, ALT: alanine aminotransferase.

**Table 3 ijms-21-04337-t003:** NITs for predicting fibrosis in nonalcoholic fatty liver disease (NAFLD).

Index	Formula	Strengths	Weaknesses
FIB-4 index [87,88,91]	(age [years] × AST [U/L]/(platelet count [10^9^/L] × √ALT [U/L]) https://www.eapharma.co.jp/medicalexpert/product/livact/fib-4/calculator.html	· Simple (only four parameters) · Accurate · Validated globally	· Requires an intermediate group · Overpredict in old patients · Inferior in patients with T2D
NAFLD fibrosis score [86]	–1.675 + 0.037 × age (years) + 0.094 × BMI (kg/m^2^) + 1.13 × impaired fasting glucose/diabetes (yes = 1, no = 0) + 0.99 × AST/ALT ratio − 0.013 × platelet count (×10^9^/L) − 0.66 × albumin (g/dL) http://nafldscore.com/	· Validated globally	· Complex (six parameters) · Requires an intermediate group · Overpredict in old patients
CA-fibrosis index [97]	1.5 × type IV collagen 7s (ng/mL) + 0.0264 × AST (IU/l)	· Simple (only two parameters)	· Only available in Japan · No external validation studies
ELF test [98]	–7.412 + (In [HA] × 0.681) + (In [P3NP] × 0.775) + (In [TIMP1] × 0.494)	· Accurate · Validated globally	· High cost (three parameters)

FIB-4: fibrosis-4, AST: aspartate aminotransferase, ALT: alanine aminotransferase, BMI: body mass index, HA: hyaluronic acid, P3NP: aminoterminal propeptide of type 3 procollagen. TIMP-1: tissue inhibitor of matrix metalloproteinase type 1, ELF: enhanced liver fibrosis.

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
