# Peer review of "Epidemiology, Pathogenesis, and Diagnostic Strategy of Diabetic Liver Disease in Japan"

_ijms, 2020, doi:10.3390/ijms21124337_

Round 1

Reviewer 1 Report

This manuscript was written about the NAFLD in relation to diabetes in Japan. It is well-written and summarized. It is better to be modified in some points.

  1. The contents in this manuscript was focusing on the diagnosis by physical and exams. If author focus on the diagnosis, it is better to change the title and abstract. Otherwise, it is better to include the therapeutic options.
  2. It is better to summarized the table about imaging findings. 
  3. It is better to make a neducational message for readers. What general clinicians take care for the early diagnosis and avoiding the disease progression?

Author Response

This manuscript was written about the NAFLD in relation to diabetes in Japan. It is well-written and summarized. It is better to be modified in some points.

1. The contents in this manuscript was focusing on the diagnosis by physical and exams. If author focus on the diagnosis, it is better to change the title and abstract. Otherwise, it is better to include the therapeutic options.

Answer: In agreement with you, we changed the abstract and the title.“Epidemiology, pathogenesis, and diagnostic strategy of diabetic liver disease in Japan“, because we focus on the diagnosis.

2. It is better to summarized the table about imaging findings.

Answer: Thank you for your kind advise. As you suggested, we summarized tables.

3. It is better to make an educational message for readers. What general clinicians take care for the early diagnosis and avoiding the disease progression?

Answer: Thank you for your kind advise. As you suggested, we added the section 9” Message to general physicians for avoiding misdiagnosis of NASH with severe fibrosis” (revised manuscript, line 400).

Reviewer 2 Report

The topic of the review is to summarize the presence of liver disease in diabetic Japan patients.
The authors went beyond the scope of the review and discussed the use of non-invasive tests for the diagnosis of NAFLD and fibrosis. They should focus on using these tests to diagnose fibrosis in
patients with diabetes. The same goes for the impact of the PNPLA3 I148M polymorphism on the risk of advanced liver injury in diabetic patients. Alternatively, the title of the review must be changed.

Major comments:
- In the Introduction (lines 35-37) the authors should rewrite the sentence “in the development of nonalcoholic fatty liver disease (NAFLD), histologically categorized as simple steatosis (nonalcoholic fatty liver: NAFL) or nonalcoholic steatohepatitis (NASH) which is not correct. “Nonalcoholic fatty liver disease (NAFLD) is histologically categorized as simple steatosis (nonalcoholic fatty liver: NAFL) and may progress to nonalcoholic steatohepatitis (NASH), characterized by hepatocytes degeneration and inflammation. The latter can advance to
fibrosis, cirrhosis and to hepatocarcinoma (HCC)”.
- Line 51: “Most Japanese people are not obese, but NAFLD is becoming more common”. NAFLD may be present also in lean subject, please discuss.
- Line 61: The prevalence of NAFLD differs by ethnic group and diagnostic tools, ranging from 15% to 70%. The worldwide prevalence of NAFLD is about 30-35%. Liver biopsy remains the “gold standard” for diagnosis of necroinflammatory activity and location of fibrosis although it is an invasive procedure. It should be mentioned in the manuscript.

Author Response

The topic of the review is to summarize the presence of liver disease in diabetic Japan patients. The authors went beyond the scope of the review and discussed the use of non-invasive tests for the diagnosis of NAFLD and fibrosis. They should focus on using these tests to diagnose fibrosis in patients with diabetes. The same goes for the impact of the PNPLA3 I148M polymorphism on the risk of advanced liver injury in diabetic patients. Alternatively, the title of the review must be changed.

Answer: In agreement with you, we changed the abstract and the title.“Epidemiology, pathogenesis, and diagnostic strategy of diabetic liver disease in Japan“, because we focus on the diagnosis.

Major comments:
- In the Introduction (lines 35-37) the authors should rewrite the sentence “in the development of nonalcoholic fatty liver disease (NAFLD), histologically categorized as simple steatosis (nonalcoholic fatty liver: NAFL) or nonalcoholic steatohepatitis (NASH) which is not correct. “Nonalcoholic fatty liver disease (NAFLD) is histologically categorized as simple steatosis (nonalcoholic fatty liver: NAFL) and may progress to nonalcoholic steatohepatitis (NASH), characterized by hepatocytes degeneration and inflammation. The latter can advance to fibrosis, cirrhosis and to hepatocellular carcinoma (HCC)”.

Answer: Thank you for your kind advise. As you suggested, we revised “NAFLD is histologically categorized as simple steatosis (nonalcoholic fatty liver: NAFL) and may progress to nonalcoholic steatohepatitis (NASH), characterized by hepatocytes degeneration and inflammation. The latter can advance to fibrosis, cirrhosis and to hepatocellular carcinoma (HCC).” (revised manuscript, line 44-47).

- Line 51: “Most Japanese people are not obese, but NAFLD is becoming more common”. NAFLD may be present also in lean subject, please discuss.

Answer: Thank you for your kind advise. As you suggested, we discussed that point (revised manuscript, line 65-68).

- Line 61: The prevalence of NAFLD differs by ethnic group and diagnostic tools, ranging from 15% to 70%. The worldwide prevalence of NAFLD is about 30-35%. Liver biopsy remains the “gold standard” for diagnosis of necroinflammatory activity and location of fibrosis although it is an invasive procedure. It should be mentioned in the manuscript.

Answer: As you suggested, the sentence “The worldwide prevalence of NAFLD is about 30-35%. Liver biopsy remains the “gold standard” for diagnosis of necroinflammatory activity and location of fibrosis although it is an invasive procedure.” Was inserted. (revised manuscript, line 84-86)

Reviewer 3 Report

The paper analyzes the influence of type 2 diabetes on liver disease in Japan. This review is complete and well written.

The Introduction has to be expanded as concern the role of oxidative stress in the liver damage of NAFLD and the interplay with cardiovascular disease.

In the paragraph 6, the value of CA-index score and the role of type IV collagen 7s have to be clarified. The Fib-4 score seems still to be the simpliest and cheapest test to define the stage of liver fibrosis

The two-step algorthm reported in Figure 1 is intriguing a and i think that could be more discussed

Author Response

The paper analyzes the influence of type 2 diabetes on liver disease in Japan. This review is complete and well written.

The Introduction has to be expanded as concern the role of oxidative stress in the liver damage of NAFLD and the interplay with cardiovascular disease.

Answer:As you suggested, we mentioned the role of oxidative stress in the liver damage of NAFLD and the interplay with cardiovascular disease (revised manuscript, line 54-57, 69-74 ).

In the paragraph 6, the value of CA-index score and the role of type IV collagen 7s have to be clarified. The Fib-4 score seems still to be the simpliest and cheapest test to define the stage of liver fibrosis.

Answer: In agreement with you, Fib-4 score is the best scoring system enough as 1st triaging tool. However, several drawbacks were explained (revised manuscript , line 264-279). For hepatologists, type IV collagen 7s or CA index are promising parameter. For general physicians, FIB-4 index is best scoring systems.

The two-step algorithm reported in Figure 1 is intriguing a and i think that could be more discussed

Answer: As you suggested, we extensively discussed (revised manuscript , line 356-382).

Round 2

Reviewer 2 Report

NA